# High-Throughput Root Imaging Analysis Reveals Wide Variation in Root Morphology of Wild Adzuki bean (*Vigna angularis*) Accessions

**DOI:** 10.3390/plants11030405

**Published:** 2022-02-01

**Authors:** Rupesh Tayade, Seong-Hoon Kim, Pooja Tripathi, Yi-Dam Choi, Jung-Beom Yoon, Yoon-Ha Kim

**Affiliations:** 1Department of Applied Biosciences, Kyungpook National University, Daegu 41566, Korea; rupesh.tayade@gmail.com (R.T.); tripathipooza21@gmail.com (P.T.); i_dont_care0@naver.com (Y.-D.C.); 2National Agrobiodiversity Center, National Institute of Agricultural Sciences, RDA, Jeonju 54874, Korea; shkim0819@korea.kr; 3Horticultural and Herbal Crop Environment Division, National Institute of Horticultural and Herbal Science, RDA, Jeonju 54874, Korea; beomi7944@korea.kr

**Keywords:** cluster analysis, image analysis, legume, root architectural trait, root morphological trait

## Abstract

Root system architecture and morphological diversification in wild accessions are important for crop improvement and productivity in adzuki beans. In this study, via analysis using 2-dimensional (2D) root imaging and WinRHIZO Pro software, we described the root traits of 61 adzuki bean accessions in their early vegetative growth stage. These accessions were chosen for study because they are used in Korea’s crop improvement programs; however, their root traits have not been sufficiently investigated. Analysis of variance revealed a significant difference between the accessions of all measured root traits. Distribution analysis demonstrated that most of the root traits followed normal distribution. The accessions showed up to a 17-fold increase in the values in contrasting accessions for the root traits. For total root length (TRL), the values ranged from 82.43 to 1435 cm, and for surface area (SA), they ranged from 12.30 to 208.39 cm^2^. The values for average diameter (AD) ranged from 0.23 to 0.56 mm. Significant differences were observed for other traits. Overall, the results showed that the accession IT 305544 had the highest TRL, SA, and number of tips (NT), whereas IT 262477 and IT 262492 showed the lowest values for TRL, SA, and AD. Principal component analysis showed an 89% variance for PC1 and PC2. K-mean clustering explained 77.4% of the variance in the data and grouped the accessions into three clusters. All six root traits had greater coefficients of variation (≥15%) among the tested accessions. Furthermore, to determine which root traits best distinguished different accessions, the correlation within our set of accessions provided trait-based ranking depending on their contribution. The identified accessions may be advantageous for the development of new crossing combinations to improve root features in adzuki beans during the early growth stage. The root traits assessed in this study could be attributes for future adzuki bean crop selection and improvement.

## 1. Introduction

Adzuki bean (*Vigna angularis*) is a traditional legume crop widely grown in East Asia, including Korea, India, China, and Japan [1]. This crop is highly valued for its high protein content, easy digestibility, and low-fat content; thus, it is globally considered a functional food for health promotion and disease prevention [2]. The genus *Vigna* comprises many wild species that belong to the *Fabaceae* family, which includes 10 domesticated species, such as cowpea, mung bean, and adzuki bean [3]. The habitats of wild *Vigna* species are so diverse that their genomes could be resources of various genes responsible for adaptation to environmental stress, which could be useful for further research in agriculture.

Climate change has led to changes in seasonal rainfall and exacerbated the prevalence of drought from spring to summer [4]. This has affected the planting season of upland crops, such as soybean, sesame, and adzuki bean [5]. Most of these crops face extended drought periods and are known to be sensitive to soil moisture content at early vegetative growth stages [6]. Thus, improving the tolerance of crops to drought stress is necessary to minimize yield loss, thereby increasing food security. Adzuki bean is one of the most widely cultivated legume crops, and it is very important to understand its root and shoot morphology that allows it to survive water-limiting conditions [6]. Studies have shown that physiological and biochemical characteristics, such as the potential activity of photosystem II and photochemical efficiency, were inhibited by drought stress in several adzuki bean plants at the early growth stage [7]. Furthermore, Chun et al. [6] reported that soil moisture content affects root development; therefore, roots are a critical factor to survival during water-limiting conditions. Although roots are important elements in soil drought, root shape has rarely been studied in adzuki bean.

The root’s primary function is to absorb water and nutrients while anchoring the plant to the soil [8,9]; at the same time, a large root system provides more support to the plant [10]. In addition, the plant’s secondary functions, such as nutrient storage, reproduction, dispersal, and synthesis of growth factors, are also supported by the roots [11]. Despite its limitations, some progress has been made in enhancing the nutrient absorption efficiency of the legume root system architecture [12,13,14]. Although roots play an essential role in plant life, there are several reasons for the scarcity of studies explicitly assessing root features. The analysis of root traits is difficult owing to expensive and inefficient facilities and technologies as well as phenotypic variations caused by biological, chemical, and physical changes in the soil [15]. Phenomics-based approaches in crops are particularly useful to collect large amounts of data of various plant growth phases using techniques and equipment that can precisely confirm relevant phenotypic traits [16]. Based on the target root trait, there are several image analysis methods available, such as X-ray imaging [17,18], magnetic resonance imaging [19], Growth and Luminescence Observatory for Roots (GLO-Roots) [20], 2D imaging [21,22], and 3D imaging [23,24,25]. As the 3D imaging method is neither feasible nor cost effective in field studies, 2D imaging techniques are increasingly being used instead [26]. Furthermore, recent advancements in high-throughput root phenotyping systems have piqued the interest of researchers and plant breeders in root trait genetic detection. In addition, due to advancements in sequencing technology many plant species genome sequences became available; several genes have been identified and their functions have been characterized [27,28]. As a result, several genes and quantitative trait loci (QTLs) have been identified and reported in a variety of plant species. This also facilitated the identification of genes or QTLs related to root system traits [29,30]. However, researchers reported relatively fewer genetic studies on root development or root system traits in plants [13,18,31,32].

The plant root system is plastic and dynamic, which allows plants to respond to their environment and optimize the acquisition of essential soil resources [33,34]. Several root traits are correlated with improved agronomic performance [35]. Optimization of the plant root system is vital to meet future demand for crop production. Recently, the agricultural production of legume crops, their development, and restrictions have been studied, and crops with adequate root systems are gaining popularity because of their improved utilization of resources and increased productivity [12,13,14]. Low resolution and low throughput limit the characterization of the root system architecture. However, thorough investigations of the morphological and architectural aspects of adzuki bean roots have not yet been conducted. The goal of this study was to characterize phenotypic variability in root attributes of 61 selected Korean adzuki bean accessions and establish selection criteria for adzuki bean genotypes with desirable root traits for efficient soil resource acquisition that can withstand adverse conditions and produce a higher yield. This experiment was performed in a greenhouse using a high throughput, image-based phenotyping method to thoroughly examine the root morphological and architectural aspects of 61 wild adzuki bean accessions that are used in Korean breeding programs for the development of enhanced adzuki bean cultivars. 

## 2. Results

### 2.1. Variability of Root Morphological Traits

The distribution curve of root traits SA, AD, NT, LAL, and LAD was normality distributed, whereas, TRL and SA were slightly skewed so for these two traits data was log-transformed and the normal distributions for all traits (TRL, SA, AD, NT, LAL, and LAD) were shown (Figure 1A–F). For all root traits, the descriptive statistics are presented in Table 1, which show the range of root morphological traits for TRL (82.43–1435 cm), SA (12.30−208.39 cm^2^), and AD (0.24−0.57 mm). Similarly, the ranges of root architectural traits NT, LAL, and LAD were 104.33−2549.20 number/plant, 0.06−0.29 cm, and 0.27−0.61 mm, respectively. Overall, significant differences were observed for all measured traits among the 61 tested adzuki bean accessions (Table 1), and the coefficient of variation (CV) was 59.37% for NT, followed by SA (56.42%), TRL (51.91%), LAL (25.45%), and AD (16.53%); the lowest variation was observed in LAD (16.16%).

The adzuki bean accessions considered in this study presented wide diversity in root traits (Figure 1). Analysis of variance (ANOVA) revealed a significant difference (*p*  <  0.0001) between the measured root traits of the 61 adzuki bean accessions (Table 2).

From the total 61 adzuki bean accessions, the top and bottom five accessions for every root morphological trait under consideration in this study were further identified and ranked (Table 3). The results showed that “IT 305544” had the highest TRL, SA, and NT, whereas IT 262477 presented the lowest value for TRL and SA. There was no matched accession between TRL and AD; thus, these two traits were shown as incompatible characters.

### 2.2. Estimation of Trait Variation

Principal component analysis (PCA) was conducted on six root traits (Table 4; Figure 2). The initial PCA on six selected traits indicated two principal components (PCs) with Eigenvalues of >1, accounting for most of the variability (89.0%) among the tested accessions. The first three PCs contributed to 98% of the total variability (Table 4). PC1 exhibited 52.2% of the overall variability, with all root traits dominating TRL and SA. PC2 was responsible for 36.9% of the total variance, dominated by the same abovementioned root traits. In addition, the angles between the arrows (Figure 2) show their approximate correlations. The numbers that are close in the plot indicate observations with similar scores on the PCA components.

### 2.3. Determination of Clusters among the Accessions Based on Root Traits

The K-means clustering analysis identified three major clusters consisting of relatively similar accessions based on root traits (Figure 3). Cluster 1 contained the lowest number of accessions (n = 6), some of which were part of the bottom ranked in AD, LAL and LAD trait measurement, such as accession 1 (IT 236774) and 2 (IT236775), whereas accession 34 (IT 262493) was bottom ranked for SA, AD, LAL, and LAD measurement. Cluster 2 grouped highest number of accessions (n = 44), 18 out 44 showed top ten ranking for TRL, SA, AD, LAL, and LAD, and another 18 with bottom ten ranking for TRL, SA, AD, LAL, and LAD, whereas the remaining 8 showed average or mixed type of values for all measured traits. Cluster 3 grouped (n = 11) accessions together, most of them ranking in the top ten for TRL, also showed higher trait values for other traits. The cluster distributions of 61 adzuki bean accessions were determined by measuring all root traits 35 days after planting (DAP) (Figure 4). Most of the accessions from cluster 3 showed higher measurement values for TRL, SA, AD, and LAD (Figure 4A–C,F) compared with those observed in the other two clusters. This suggests that the accessions from cluster 3 are distinct from the others and have better root traits, which may allow for good adaption, nutrient uptake, and plant performance during exposure to biotic and abiotic stresses and can be considered for future crop improvement research. 

### 2.4. Correlation among Root Traits

To identify the possible relationships between the measured root traits, all six root traits were selected for Pearson’s correlation analysis. There was a strong correlation between morphological (TRL, SA, AD) and architectural traits (NT, LAL, and LAD) (Figure 5). Root length-related traits, such as TRL and SA, showed the highest correlation values compared with other traits. TRL showed a strong positive correlation with SA (r^2^ = 0.93 ***) at the significance level (*p* ≤ 0.0001). Similarly, NT was positively correlated with TRL (r^2^ = 0.83 ***) and SA (r^2^ = 0.67 ***). In contrast, LAL (r^2^ = −0.51 ***) and AD (r^2^ = −0.31 ***) showed negative correlations with NT at *p* ≤ 0.0001. Therefore, TRL showed the highest positive correlation with SA within root morphological traits, and NT revealed the highest negative correlation with LAL within root architectural traits. In contrast, AD presented the highest correlation with LAD (r^2^ = 0.93 ***) (Figure 5).

## 3. Discussion

### 3.1. Characterization of Wild Adzuki Beans Based on Root Traits and Importance for Crop Improvement

As root morphological and architectural traits are crucial for crop development and yield, researchers are showing a great interest in studying root phenotypes in different plant species for breeding strategies [36,37]. Root phenotyping has been conducted in several crops, such as soybean [22,38], rapeseed [39], and rice [24]. Furthermore, previous studies on diverse plants have suggested that root traits are the most significant for the acquisition of edaphic resources, such as water and nutrients [40,41,42,43]. Moreover, plants with varying genetic and nutritional conditions exhibit extensive variations in root system distribution, which allows plants to recover faster and survive under stress conditions [44]. Adzuki bean selection and crop improvement programs largely emphasize above-ground agronomic and yield traits, such as height, leaf size, leaf SA, number of pods, and seed weight, whereas the impact of root attributes are mostly ignored [45,46]. Therefore, in this study, we investigated the root traits of 61 wild adzuki bean accessions used in Korean breeding programs to better understand root system features. The characterization of these accessions can facilitate a more rapid application of root phenotypes in the breeding of new adzuki bean cultivars.

### 3.2. Advantages of Root Traits Variation among the Adzuki Bean Accessions 

This study analyzed various root traits, such as TRL, SA, AD, NT, LAL, and LAD, at the early growth stage using a polyvinyl chloride (PVC) pipe, soil-based experiment method and a 2D high-throughput phenotyping system [22,47]. Root phenotyping results showed a significant difference in TRL, SA, AD, NT, LAL, and LAD values among the tested adzuki bean accessions at *p* ≤ 0.0001 (Table 2). Further, the CV suggested the presence of wide phenotypic variation among the measured traits of the tested accessions. Specifically, large variation was shown for NT, which is considered to be involved in stress signaling for the modification of phytohormone level. In this context, a study reported that upon the onset of drought stress, ABA content in the root tips increased constantly by nearly four times every hour, which implies that these plant parts have an important role in signaling a stress response [48]. The accessions used in this study presented a desirable variation in root traits that can be used in new crossing combinations to create more diverse root characteristics in adzuki bean and also to develop new cultivars. 

In the present study, TRL presented a strong positive correlation with SA and NT and a negative one with AD. Previous studies reported a similar trend in root phenotyping of other legume crops, such as soybean [22,49]. In addition, the association of root traits (such as TRL) with root mass and root depth, which leads to better adaptation and nutrient uptake in different plant species (including legume species), has also been reported [13,32,50]. Furthermore, longer roots with fine diameters are strongly correlated with stomatal conductance of water vapor, ultimately supporting the water retention capacity of plants [51]. Similarly, a higher root mass at a greater soil depth has been reported to extract more water from deeper soil profiles; however, an increase in root biomass has not been associated with increased water uptake [52,53,54]. The current study revealed that accession clustering patterns were most likely caused by variation in root traits, and grouping of accessions with contrasting root traits was also observed. The clustering of genotypes from the same category into different clusters has been previously reported in sorghum [55]. Overall, root phenotyping results showed that the tested wild adzuki bean accessions had a wide variation for the measured root traits, and there was a certain correlation among them. Therefore, the root traits measured in this study can be used for the future selection of adzuki bean crops characterized by a better ability to adapt to abiotic and biotic stress conditions and enhanced nutrient uptake and yield. However, there is a limitation in employing phenotyping results from early growth to harvesting stages in breeding programs. Thus, further studies should investigate root phenotyping relationships in both controlled and field conditions at different growth stages. Phenotyping systems should ensure reliable evaluations of root traits, which can be utilized at any stage of crop development. 

## 4. Materials and Methods

### 4.1. Plant Materials and Growth Conditions

In this experiment, 61 wild adzuki bean accessions were used (Appendix A). The experiment was set up as a completely randomized design with five replications. Seeds were sown in pots created out of PVC pipes with a diameter of 6 cm and height of 40 cm, containing horticultural soil (Tobirang, Baekkwang Fertility, Korea). Seed scarification was performed before planting, which was carried out on 28 May 2021, and the harvest took place on 1 July 2021. The plants were grown in a greenhouse located at the Kyungpook National University Research Center, Daegu, Korea. Two scarified seeds were planted in each pot; then, we retained a plant for root analysis. During the experiment, plants were irrigated regularly, and greenhouse conditions were maintained at an average temperature of 32 ± 3 °C and humidity of 67% ± 5%. 

### 4.2. Phenotypic Data Collection

All plants were harvested 35 DAP at the second or third trifoliate leaf stage. To collect root samples, a sieve was prepared, and the PVC pipe was poured into it; then, the whole plant samples were carefully removed from the dumped soil. The shoot parts were removed from the plant samples with scissors, and only the root parts were used for analysis. The collected roots were washed in clean tap water to remove foreign substances and soil particles. To prevent them from drying, the samples were placed in a zip-lock bag containing 10−15 mL of water that was stored in an icebox, and they were refrigerated until root analysis was performed. The roots were spread out on a transparent acrylic tray (30 × 20 cm) and submerged in clean water for scanning. They were scanned using an Expression 12000XL scanner (Epson, Suwa, Nagano, Japan) to acquire 2D root images. These were analyzed using the WinRHIZO Pro software (Reagent Instruments, Quebec, Canada). Six parameters were used in this experiment: TRL, SA, AD, NT, LAL, and LAD. Particularly, LAL is a root part between two forks or a fork and a tip. It is basically a study of the morphology and basic connectivity of roots segments measured by average length of links that belong to the order. Similarly, LAD is the average diameter for links that belongs to order; link analysis was measured by Regent’s unique method and with Tennant’s statistical method. Root overlap at forks and tips are considered to provide accurate measurements of length and area (https://regent.qc.ca/assets/winrhizo_software.html Access on 12 October 2021). The measurement and recording of each numerical value on the images were determined within the range set during analysis, and each category was defined as root TRL, root SA, root AD, number of root ends, and root average in link units.

### 4.3. Statistical Analysis

ANOVA was performed to determine statistical significance, and a histogram was created; descriptive statistics were generated using SAS release 9.4 (SAS, Gary, NC, USA). Correlation analysis, PCA, and K-mean cluster analysis were conducted using R studio (version 4.0.5). Figures were created in Microsoft Excel (2013) and R studio (version 4.0.5) to evaluate the root morphological traits of the adzuki bean accessions.

## 5. Conclusions

In this study, 61 wild adzuki bean accessions were characterized at the early growth stage, and their root morphological and architectural traits were determined using a 2D high-throughput system. Large variation between the accessions was observed in all root traits, and the varying degrees of these relationships were determined. Furthermore, this study identified the accessions with the highest and lowest root trait values, which could be useful for the creation of new crossing combinations with improved root traits in adzuki bean at the early growth stage. Additional research may reveal promising traits for future adzuki bean crop selection and improvement, leading to a greater adaptation to abiotic and biotic stresses, as well as an increased nutrient uptake and yield.

## Figures and Tables

**Figure 1 plants-11-00405-f001:**
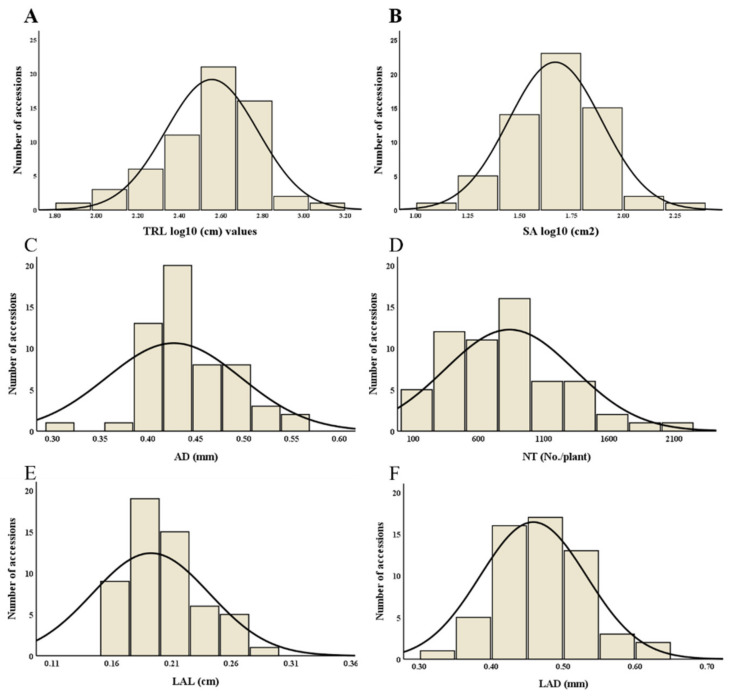
Histogram of normal distribution curves for root morphological traits. In the figure, each abbreviation indicates (**A**) total root length (TRL), (**B**) surface area (SA), (**C**) average diameter (AD), (**D**) number of tips (NT), (**E**) link average length (LAL), and (**F**) link average diameter (LAD).

**Figure 2 plants-11-00405-f002:**
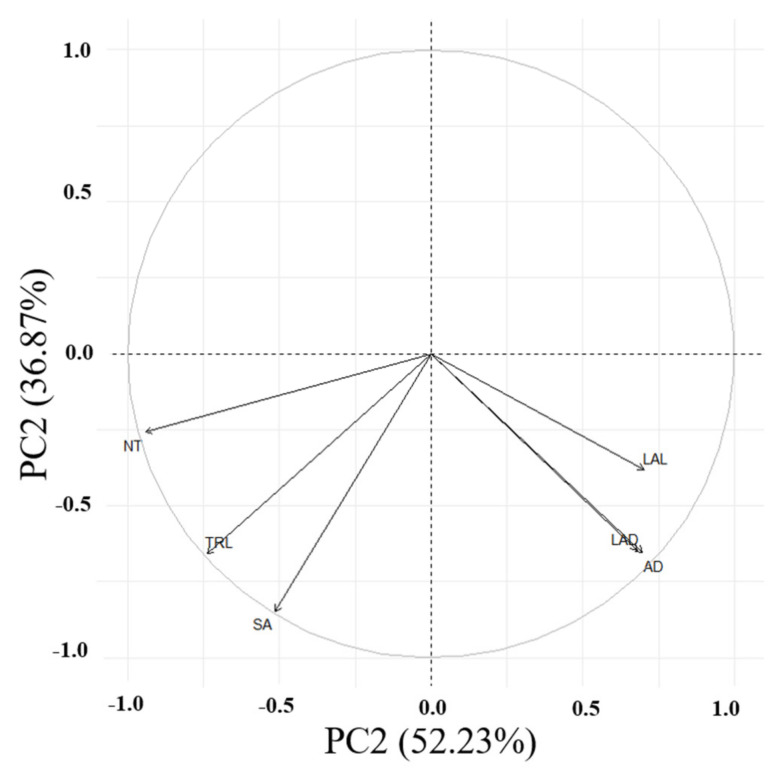
Principal component analysis of six root traits for the adzuki bean accessions. PC1 vs. PC2 represents 89.10% of the total variation. Abbreviations: TRL, total root length; SA, surface area; AD, average diameter; NT, number of tips; LAL, link average length; LAD, link average diameter.

**Figure 3 plants-11-00405-f003:**
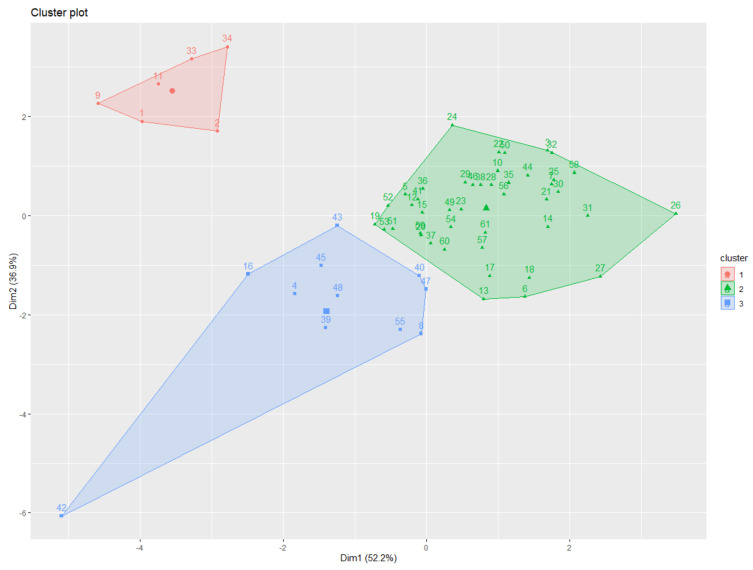
Visualization of the spatial relationships between adzuki bean accession clusters formed by plotting the first two dimensions. Dots with different colors indicate comparable accessions gathered together in a cluster. The results of K-means clustering (three clusters) using the first two principal component analysis dimensions are shown using different colors representing cluster distributions.

**Figure 4 plants-11-00405-f004:**
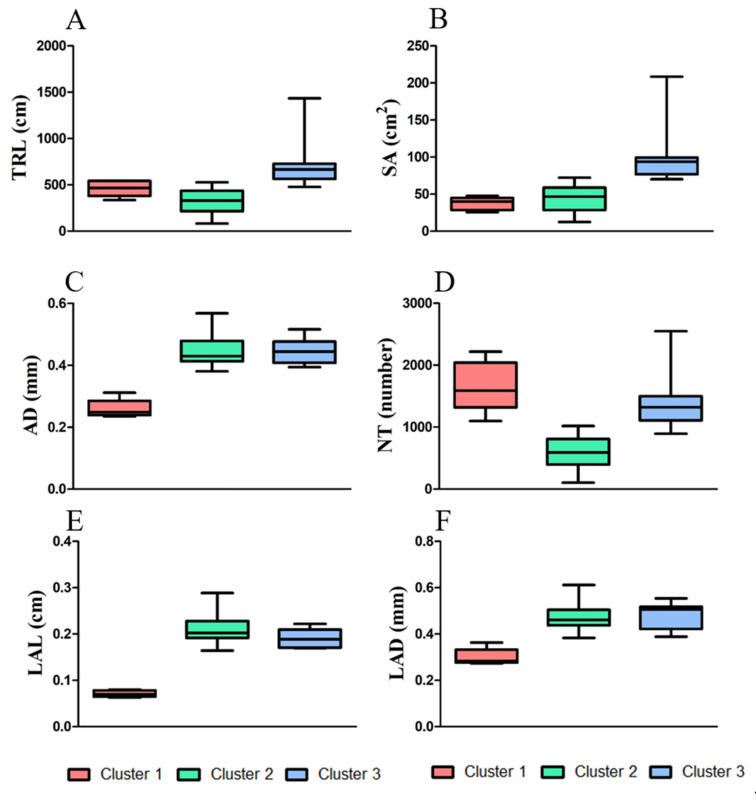
Box plots displaying the measured root traits for individual clusters, and according to the number of accessions belonging to each cluster. The central value and variability of the 61 accessions are based on the three clustering groups obtained from K-means cluster analysis. The six root traits were plotted as follows: (**A**) TRL, total root length; (**B**) SA, surface area; (**C**) AD, average diameter; (**D**) NT, number of tips; (**E**) LAL, link average length; and (**F**) LAD, link average diameter.

**Figure 5 plants-11-00405-f005:**
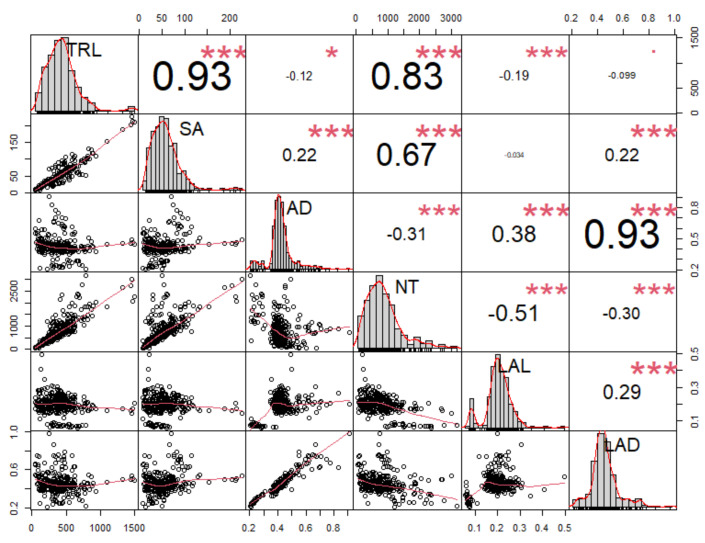
Correlation analysis between six root traits (TRL, SA, AD, NT, LAL, and LAD) in 61 adzuki bean wild accessions examined 35 days after planting at the second or third trifoliate leaf stage. Abbreviations: TRL, total root length; SA, surface area; AD, average diameter; NT, number of tips; LAL, link average length; LAD, link average diameter. In the figure, asterisks (*) represent significance at the 0.05 level; *p* ≤ 0.01 (*), *p* ≤ 0.001 (**), and *p* ≤ 0.0001 (***).

**Table 1 plants-11-00405-t001:** Descriptive statistics for six root traits in 61 adzuki bean accessions.

Traits	Range	Mean	SD ^a^	CV (%) ^b^	Skewness	Kurtosis
TRL	82.43–1435	406.03	210.77	51.91	–0.37 ^c^	0.56 ^c^
SA	12.30–208.39	53.52	30.20	56.42	–0.06 ^c^	0.39 ^c^
AD	0.24–0.57	0.43	0.07	16.53	−0.96	1.69
NT	104.33–2549.20	836.47	496.65	59.37	1.27	1.98
LAL	0.06–0.29	0.19	0.05	25.45	−1.17	1.82
LAD	0.27–0.61	0.46	0.07	16.16	−0.53	0.85

^a^ Standard deviation; ^b^ Coefficient of variation in percentage; ^c^ log10 values.

**Table 2 plants-11-00405-t002:** Analysis of variance of six root morphological traits.

Traits	Source	DF	Type III SS	Mean Square	F Value	*Pr > F*
TRL	accessions	60	12785904	213098.4	31.08	<0.0001
	rep	4	1741380	435345	63.49	<0.0001
SA	accessions	60	261505.8	4358.43	27.34	<0.0001
	rep	4	26277.92	6569.48	41.2	<0.0001
AD	accessions	60	1.385495	0.023092	4.29	<0.0001
	rep	4	0.072648	0.018162	3.38	0.0104
NT	accessions	60	70466504	1174442	17.27	<0.0001
	rep	4	5495104	1373776	20.21	<0.0001
LAL	accessions	60	0.679572	0.011326	9.38	<0.0001
	rep	4	0.012475	0.003119	2.58	0.038
LAD	accessions	60	1.534246	0.025571	3.49	<0.0001
	rep	4	0.058615	0.014654	2.00	0.0955

Abbreviations: TRL, total root length; SA, surface area; AD, average diameter; NT, number of tips; LAL, link average length; LAD, link average diameter.

**Table 3 plants-11-00405-t003:** List of highest and lowest-ranking accessions for each root trait.

Top 10 Accession for Each Trait			Bottom 10 Accession for Each Trait		
Rank	Accession Name	TRL (cm)	Rank	Accession Name	TRL (cm)
1	IT 305544	1434.99	1	IT 305597	204.81
2	IT 262454	766.77	2	IT 262471	203.51
3	IT 262413	727.90	3	IT 262469	190.17
4	IT 262500	693.75	4	IT 262485	185.71
5	IT 305602	677.41	5	IT 262472	185.07
6	IT 305594	665.51	6	IT 262484	175.84
7	IT 305588	654.45	7	IT 242846	138.09
8	IT 262424	608.63	8	IT 305605	135.17
9	IT 305585	564.99	9	IT 262487	131.64
10	IT 262501	549.23	10	IT 262477	82.43
**Rank**	**Accession Name**	**SA (cm^2^)**	**Rank**	**Accession Name**	**SA (cm^2^)**
1	IT 305544	208.39	1	IT 262484	26.07
2	IT 262500	103.61	2	IT 262472	26.05
3	IT 305602	99.37	3	IT 305597	25.66
4	IT 262413	97.40	4	IT 262493	25.62
5	IT 262454	96.02	5	IT 262471	24.26
6	IT 262424	93.55	6	IT 262469	24.01
7	IT 305594	90.91	7	IT 305605	19.09
8	IT 305588	83.15	8	IT 242846	18.86
9	IT 305591	76.75	9	IT 262487	17.61
10	IT 262501	74.88	10	IT 262477	12.29
**Rank**	**Accession Name**	**AD (mm)**	**Rank**	**Accession Name**	**AD (mm)**
1	IT 262422	0.57	1	IT 262464	0.39
2	IT 262457	0.57	2	IT 305599	0.39
3	IT 262485	0.53	3	IT 262417	0.39
4	IT 262446	0.52	4	IT 262471	0.38
5	IT305591	0.52	5	IT 236775	0.31
6	IT 262478	0.51	6	IT 236774	0.28
7	IT 262477	0.51	7	IT 262443	0.25
8	IT 262424	0.50	8	IT 262434	0.25
9	IT 262451	0.49	9	IT 262493	0.24
10	IT 262456	0.49	10	IT 262492	0.24
**Rank**	**Accession Name**	**NT (number)**	**Rank**	**Accession Name**	**NT (number)**
1	IT 305544	2549.20	1	IT 305597	388.00
2	IT 262434	2218.00	2	IT 305586	385.80
3	IT 236774	1983.20	3	IT 262423	365.00
4	IT 262443	1739.00	4	IT 262485	352.80
5	IT 262454	1576.00	5	IT 262484	318.60
6	IT 262500	1499.00	6	IT 262487	247.40
7	IT 236775	1437.25	7	IT 262472	233.67
8	IT 262413	1393.20	8	IT 242846	199.60
9	IT 262492	1392.80	9	IT 305605	196.00
10	IT 305588	1337.00	10	IT 262477	104.33
**Rank**	**Accession Name**	**LAL (cm)**	**Rank**	**Accession Name**	**LAL (cm)**
1	IT 262423	0.29	1	IT 305544	0.17
2	IT 262478	0.27	2	IT 262441	0.17
3	IT 262466	0.27	3	IT 262499	0.17
4	IT 305603	0.26	4	IT 262498	0.16
5	IT 305608	0.26	5	IT 262492	0.08
6	IT 305601	0.25	6	IT 262493	0.08
7	IT 262477	0.25	7	IT 262443	0.07
8	IT 262464	0.23	8	IT 262434	0.07
9	IT 305586	0.23	9	IT 236774	0.07
10	IT 262470	0.23	10	IT 236775	0.06
**Rank**	**Accession Name**	**LAD (mm)**	**Rank**	**Accession Name**	**LAD (mm)**
1	IT 262477	0.61	1	IT 262464	0.40
2	IT 262478	0.60	2	IT 262417	0.39
3	IT 262422	0.60	3	IT 262454	0.39
4	IT 262446	0.58	4	IT 262471	0.38
5	IT 262424	0.55	5	IT 236775	0.36
6	IT 262451	0.55	6	IT 236774	0.32
7	IT 262456	0.54	7	IT 262443	0.28
8	IT 262485	0.53	8	IT 262434	0.28
9	IT 262484	0.53	9	IT 262493	0.28
10	IT 305604	0.53	10	IT 262492	0.27

Abbreviations: TRL, total root length; SA, surface area; AD, average diameter; NT, number of tips; LAL, link average length; LAD, link average diameter.

**Table 4 plants-11-00405-t004:** Variable loading score of six root traits and variance proportion for each principal component.

Traits	Principal Component (Eigenvectors)		
	1	2	3
TRL	0.74	0.66	0.13
SA	0.51	0.85	0.08
AD	−0.70	0.66	−0.24
NT	0.94	0.26	−0.10
LAL	−0.70	0.38	0.59
LAD	−0.68	0.65	−0.30
Proportion of Variation			
Eigenvalues	3.13	2.21	0.54
Variance (%)	52.23	36.87	8.92
Cumulative (%)	52.23	89.10	98.02

Abbreviations: TRL, total root length; SA, surface area; AD, average diameter; NT, number of tips; LAL, link average length; and LAD, link average diameter.

## Data Availability

Not applicable.

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
