# Peer review of "High-Throughput Root Imaging Analysis Reveals Wide Variation in Root Morphology of Wild Adzuki bean (Vigna angularis) Accessions"

_plants, 2022, doi:10.3390/plants11030405_

Round 1

Reviewer 1 Report

Lines 22 and 23 of the abstract states that "the distribution analysis showed that all root traits followed a normal distribution", however, the results state that "TRL and SA values in the analysed accessions were slightly skewed with right tails heavier than those observed in the normal distribution", (Line 124-126). If the variables TRL and SA do not have a normal distribution, a parametric test such as ANOVA cannot be applied. Try transforming the data, e.g. with the logarithm, or apply a non-parametric test.

Table 1 shows that there are statistically significant differences in the replicates of the experiments. This result could partly invalidate the statistical analyses performed.

Line 70-71. The main function of the root is not to overcome the uneven distribution of nutrients in the soil, but the uptake of water and nutrients and the anchoring of the plant to the soil.

Line 80-81. “As a result, current research relies on phenomics analyses “... there are also many genetic studies. Please include further bibliography.

Line 86-88. Check this paragraph and the bibliographic citation.

Line 99-107. This paragraph is irrelevant. WinRHIZO is a programme that has been in use for years and is well known and used in the study of the root system.

In the Introduction, clearly define the objectives of the study.

Has intra-accession variability been observed? If so, this should be indicated in the results and taken into account in the differentiation between accessions and also in the Discussion section.

ANOVA results (Lines118-120) should be placed in the text after the descriptive statistics.

Line 149. Replace incompatible characters with independent characters.

Line 162-163. This is an obvious comment.

Please, improve the quality of Figure 2.

Line 172-178. There are discrepancies between the clusters described in the text and what is shown in Figure 3.

Line 178-180. Rewrite for clarity

Line 239. The LAL trait also shows significant differences.

Line 250-251. “which implies that most adzuki bean accessions with a high TRL had a lower AD and vice versa”, this is obvious.

Line 257-262. I do not understand the premise, as the authors do not assess the drought tolerance of the accessions.

Line 264-266. Rewrite for clarity

In the discussion, an attempt should be made to explain what could be the origin of the three clusters of accessions. What do the clusters show? For example, if the samples come from neighbouring areas, or if they grow in the same type of soil, climate, etc.

Line 301. Describe what the LAL and LAD variables are and how they are measured.

Author Response

Lines 22 and 23 of the abstract states that "the distribution analysis showed that all root traits followed a normal distribution", however, the results state that "TRL and SA values in the analysed accessions were slightly skewed with right tails heavier than those observed in the normal distribution", (Line 124-126). If the variables TRL and SA do not have a normal distribution, a parametric test such as ANOVA cannot be applied. Try transforming the data, e.g. with the logarithm, or apply a non-parametric test.

Answer: Thank you for commenting on it, and also suggesting the valuable clue, we transformed the data and updated the information based on the transformed data for TRL and SA traits.

Table 1 shows that there are statistically significant differences in the replicates of the experiments. This result could partly invalidate the statistical analyses performed.

Answer: We agree with the worthy reviewer and understand the point raised but, the data was obtained as it is and we presented the same. This study was conducted with replications to analyze ANOVA. This ANOVA analysis considers removing the block to block variation from the experimental error to increase the precision of this analysis. We assume this significance for replication effect is not the issue for this analysis and hope the reviewer kindly considers our point. Secondly, WinRHIZO is very sensitive and little background noise might have also contributed to TRL and SA variation.

Line 70-71. The main function of the root is not to overcome the uneven distribution of nutrients in the soil, but the uptake of water and nutrients and the anchoring of the plant to the soil.

Answer: Thank you for commenting on it, we have modified the sentence as below.

“The root's primary function is to absorb water and nutrients while anchoring the plant to the soil”

Line 80-81. “As a result, current research relies on phenomics analyses “... there are also many genetic studies. Please include further bibliography.

Answer: Thank you for commenting on it, we have removed the sentence and provided the genetic studies reported in plants (L98-109).

Line 86-88. Check this paragraph and the bibliographic citation.

Answer: Thank you for the suggestion, now we have provided the references for mentioned technology and modified bibliographic citation.

Line 99-107. This paragraph is irrelevant. WinRHIZO is a programme that has been in use for years and is well known and used in the study of the root system.

Answer: Sorry for the inconvenience we have removed this paragraph.

In the Introduction, clearly define the objectives of the study.

Answer: We have modified the introduction and clearly defined the objective of the study as b mentioned below.

The goal of this study was to characterize phenotypic variability in root attributes of selected 61 Korean adzuki bean accessions and establish selection criteria for adzuki bean genotypes with desirable root traits for efficient soil resource acquisition and can withstand adverse conditions and produce a higher yield.”

Has intra-accession variability been observed? If so, this should be indicated in the results and taken into account in the differentiation between accessions and also in the Discussion section.

Answer: We do not observe the intra-accession variability, maybe due to background noise we found a noticeable difference in replication for three traits but the wild accessions are assumed to be mostly fixed or homozygous for the root traits. To avoid further ambiguity we have not elaborated this in the MS.

ANOVA results (Lines118-120) should be placed in the text after the descriptive statistics.

Answer: Thank you for the suggestion now we shifted the content as per the suggestion.

Line 149. Replace incompatible characters with independent characters.

Answer: We have corrected it, thank you for the suggestion.

Line 162-163. This is an obvious comment.

Answer: We have modified the sentence.

Please, improve the quality of Figure 2.

Answer: Thank you for the suggestion we have replaced with an improved image.

Line 172-178. There are discrepancies between the clusters described in the text and what is shown in Figure 3.

Answer: Sorry for the inconvenience we have revised the content and corrected the mistake.

Line 178-180. Rewrite for clarity

Answer: Sorry for the inconvenience we have revised the sentence as mentioned below.

The cluster distributions of 61 adzuki bean accessions were determined by measuring all root traits 35 days after planting (DAP) (Figure 4).”

Line 239. The LAL trait also shows significant differences.

Answer:  Thank you for commenting on it, now we have included LAL results in the sentence.

Line 250-251. “which implies that most adzuki bean accessions with a high TRL had a lower AD and vice versa”, this is obvious.

Answer: We removed the mentioned sentence. 

Line 257-262. I do not understand the premise, as the authors do not assess the drought tolerance of the accessions.

Answer:  We removed it from the discussion part.

Line 264-266. Rewrite for clarity

Answer: Sorry for the inconvenience we have revised the sentence as mentioned below.

In the discussion, an attempt should be made to explain what could be the origin of the three clusters of accessions. What do the clusters show? For example, if the samples come from neighbouring areas, or if they grow in the same type of soil, climate, etc.

Answer: Thank you for the suggestion but these accessions are collected a quite similar region from South Korea and not other parts of the world so the origin of accessions is clear and complete characterization with soil type, the climate is not available, hence we are unable to discuss cluster further due to lack of information. However, we have discussed it based on the root trait variation.

Line 301. Describe what the LAL and LAD variables are and how they are measured.

Answer:  Thank you for the valuable suggestion we have incorporated the required information.

Answer: We would like to thank the worthy reviewers for the time given to this manuscript. All the comments and suggestions were genuine, valuable, and improved the quality of the manuscript. We highly appreciate your efforts and agree to the suggested changes. All the changes that were made in the revised MS can be found with the track change red clolour for Reviwer1 comments and blue for Reviwer2 comments.

Reviewer 2 Report

This study is  well written and provides a good results presentation, a complete discussion and good the presentation of the materials and methods.

The only relevant comment I've got is that In Table 3 there is a problem in the second half (NT, LAL et LAD values), values are repeated in the top varieties are equal to the botton ones, see file attached.

The legends of two figures should be completed with the abbreviations used

Author Response

Reviewer 2

This study is  well written and provides a good results presentation, a complete discussion and good the presentation of the materials and methods.

The only relevant comment I've got is that In Table 3 there is a problem in the second half (NT, LAL et LAD values), values are repeated in the top varieties are equal to the botton ones, see file attached.

The legends of two figures should be completed with the abbreviations used

Answer: We would like to thank the worthy reviewer for the time he given to this manuscript. We have made the suggested correction and revise the content. All the changes that were made in the revised MS can be found with the track change red clolour for Reviwer1 comments and blue for Reviwer2 comments.

Round 2

Reviewer 1 Report

Please, indicate in the Results section that the TRL and SA variables were not normally distributed, so they were transformed. Please, indicate the type of transformation performed.

Author Response

Please, indicate in the Results section that the TRL and SA variables were not normally distributed, so they were transformed. Please, indicate the type of transformation performed.

       Answer: Based on your comments, we mentioned it line 136 - 139. It was marked as blue-green color. Thanks.